# Condition Monitoring of Active Magnetic Bearings on the Internet of Things †

**Alexander H. Pesch \* and Peter N. Scavelli**

Department of Engineering, Hofstra University, 104 Weed Hall, Hempstead, NY 11549, USA; PScavelli1@Pride.Hofstra.edu

\* Correspondence: Alexander.H.Pesch@Hofstra.edu; Tel.: +1-516-463-7175

† This paper is an extended version of our paper published in: Pesch, A.H.; Scavelli, P.N. Condition Monitoring of AMBs on the IoT. In Proceedings of the 16th International Symposium on Magnetic Bearings (ISMB16), Beijing, China, 13–17 August 2018.

**Abstract:** A magnetic bearing is an industrial device that supports a rotating shaft with a magnetic field. Magnetic bearings have advantages such as high efficiency, low maintenance, and no lubrication. Active magnetic bearings (AMBs) use electromagnets with actively controlled coil currents based on rotor position monitored by sensors integral to the AMB. AMBs are apt to the Internet of Things (IoT) due to their inherent sensors and actuators. The IoT is the interconnection of physical devices that enables them to send and receive data over the Internet. IoT technology has recently rapidly increased and is being applied to industrial devices. This study developed a method for the condition monitoring of AMB systems online using off-the-shelf IoT technology. Because off-the-shelf IoT solutions were utilized, the developed method is cost-effective and can be implemented on existing AMB systems. In this study, a MBC500 AMB test rig was outfitted with a Raspberry Pi single board computer. The Raspberry Pi monitors the AMB's position sensors and current sensors via an analog-to-digital converter. Several loading cases were imposed on the experimental test rig and diagnosed remotely using virtual network computing. It was found that remote AMB condition monitoring is feasible for less than USD 100.

**Keywords:** Active Magnetic Bearing; AMB; Internet of Things; IoT; Condition Monitoring

## 1. Introduction

Active Magnetic Bearings (AMBs) support a rotor with a magnetic field such that the rotor is levitated [1]. AMBs are an alternative to other types of bearings such as rolling element bearings and fluid film bearings [2]. AMBs have advantages over other types of bearings such as the potential for higher efficiency and rotational speeds. In addition, AMBs do not need lubrication, which is advantageous in clean rooms, and food or medical processing (e.g., [3]). AMBs do not need service and are useful in subsea [4] and space applications [5]. The control in the airgap clearance can be exploited, for example, for machining tool positioning [6] and active balancing [7].

Passive magnetic bearings use permanent magnet stators to repel a rotor with permanent magnets with opposing polarization [8]. The system is naturally stable, tending toward the equilibrium in the center of the bearing. Passive magnetic bearings tend to be more efficient than AMBs with electromagnets, because there is no current consumption. However, they lack the load capacity and performance of the actuated AMBs. Permanent-magnet-biased AMBs have permanent magnets to provide initial pulling force on a ferromagnetic rotor. Permanent-magnet-biased AMBs serve as a way to achieve some of the advantages of passive magnetic bearings and AMBs and still utilize integral

sensors and actuators [9]. Zero-bias and low-bias AMBs also use integral sensors and high-efficiency actuators and require sophisticated control laws (e.g., [10–16]) to take advantage of the nonlinear flux.

AMBs use electromagnetic actuators to generate an attractive force on a ferromagnetic rotor. The magnitude of the attractive force increases as the rotor moves closer to the stator. Therefore, the setup is naturally unstable and requires stabilizing feedback control in order to function. An AMB inherently includes some form of position sensing. The rotor position signal is used to calculate required coil current to control rotor position and achieve stable levitation. A typical type of sensor is the noncontact eddy current position probe. AMB position sensors have conveniently been utilized for online health monitoring [17]. In addition, real-time knowledge of rotor position and coil current can be used to determine bearing forces that can be used to indirectly monitor rotor loading [18].

AMBs are designed, built, and implemented by engineers and technicians with expert knowledge. For example, AMB controller design must be customized to rotor geometry because of inertial and gyroscopic cross-coupling and to avoid excitation of flexible modes. Once commissioned on-site, the AMB can be used by the end user with relatively little training [19]. However, the end user may not be able to effectively troubleshoot complications with the AMB system that may arise after the commissioning process, because the end users are not trained to recognize or diagnose these complications. In such cases, a field service technician from the AMB's Original Equipment Manufacturer (OEM) must go on-site to perform service. This results in down time for the system supported by the AMB and increased expense to the customer. A solution is to make AMBs part of the Internet of Things (IoT). This would allow for increased productivity and decreased costs.

The exact definition and scope of the IoT is still being developed. However, the IoT basically enables the interconnection of physical devices. This interconnection allows the devices to send and receive data over the Internet. This enables value creation beyond the mere sum of the "thing-based function" and "IT-based service" [20]. By putting AMBs on the IoT, a remote user such as an off-site OEM technician could access the AMB, and diagnose malfunctions without the need for on-site examination. Therefore, (to reduce or eliminate equipment down time) the OEM technician can recommend corrective action immediately, or even preemptively.

Early cases of what would become known as IoT were in the area of radio-frequency identification (RFID) tags (e.g., [21]). Since then, there has been much development of IoT because of the significant impact on people's everyday lives [22]. There are several instances of industry beginning to take advantage of IoT technology [23]. More recently, IoT has been applied towards structural health monitoring [24]. For example, IoT has been used for monitoring the position of steel in a continuous casting process [25]. In addition, IoT has been used for the monitoring of vibrations in electric motors [26]. This suggests the potential of AMBs when coupled with the IoT.

There has been some previous work in the area of IoT tools used for AMBs. These are mostly proprietary industrial systems used toward facilitating automated commissioning. However, there has not been enough work in the application of off-the-shelf IoT hardware, which is low-cost and readily available. An early work in remote operation of AMBs is found in [27], where a local area network (LAN) is used to facilitate communication with real-time AMB controllers in a laboratory environment. The utilized LAN is a direct connection between the remote computer used for system interfacing and computers on the AMBs with dedicated hardware for real-time control and ethernet communications. This method is successful at interfacing with the AMBs for conducting experiments at a safe distance but was hardware intensive. In [28], a remote computer is used to communicate with a server computer via Transmission Control Protocol/Internet Protocol (TCP/IP). The server passes data via an RS-232 connection with a digital signal processor, which in turn controls an AMB system. The setup is used to remotely tune the AMB controller gains. Jayawant and Davies [29] developed an automated commissioning scheme capable of remote commissioning AMBs via TCP/IP and a Simple Object Access Protocol (SOAP) interface. SOAP sends packaged datasets between computers on a high level. Because the data are compiled and packaged before being transmitted via SOAP, the method can facilitate data transfer between different types of systems, e.g., differing operating

systems. With SOAP, the actual data vector from the AMB is passed between the local and remote computers. Data processing can take place on either or both computers. In [30], SOAP-based remote commissioning is applied to a fluid film bearing AMB test rig and an industrial turbo-machine. Similar remote commissioning methods are utilized in [31] for a 3.3 MW motor-driven compressor and in [32] for a high-temperature gas-cooled reactor.

In the present study, an AMB test rig was augmented with an off-the-shelf IoT gateway that is low-cost and readily available. The local device was programmed to read the AMB's sensors and perform data processing for condition monitoring. A remote user could then log into the device through Virtual Network Computing (VNC) via TCP/IP to observe the sensor signals. This approach differs from the SOAP approach in that the AMB data vectors are not transmitted to the remote computer. In the current approach, all data processing is done on the local IoT gateway and the resulting frames are used remotely for condition monitoring. The usefulness of the developed system for condition monitoring of AMBs was demonstrated by operating the test rig under different conditions, and presenting the remote user interface, illustrating how the condition of the system was evaluated. Preliminary results for this study are presented in [33].

The next section explains the concept of using the IoT for condition monitoring of AMBs. Then, the experimental system used to demonstrate the proposed method is detailed, including cost information. Next, the experimental results are presented. The practical issue of sampling time when utilizing the IoT gateway device is then discussed. Finally, the paper is ended with concluding remarks.

## 2. Materials and Methods

This section covers the proposed method and the materials used to implement the experimental demonstration. Specifically, an introduction to AMBs and the proposed method for condition monitoring of AMBs on the IoT is discussed in the next subsection. Then, the AMB test rig for the experimental demonstration is presented. Finally, subsections for the hardware and software for IoT implementation are presented.

### 2.1. AMB and Condition Monitoring on the IoT

An AMB uses a magnetic field to support a rotating shaft. The magnetic field is generated by an array of electromagnets around the rotor. The electromagnetic force induced on the ferromagnetic rotor is inherently unstable. The position of the shaft is measured in real time by a noncontact position sensor. The position data are used (by a controller) to calculate how much current is needed in the electromagnetic coils to maintain a stable levitation. A typical control setup for AMBs is shown in Figure 1 [34]. Figure 1a shows a common biasing scheme for a single AMB axis. Figure 1b shows a control block diagram for a generic AMB-rotor system, which is Multiple-Input-Multiple-Output (MIMO) because a rotor may have multiple AMBs. Therefore, an AMB, by its nature, includes sensors and actuators. It is suitable for condition monitoring for traditional rotordynamic faults [35]. It is also highly apt to be extended to the IoT.

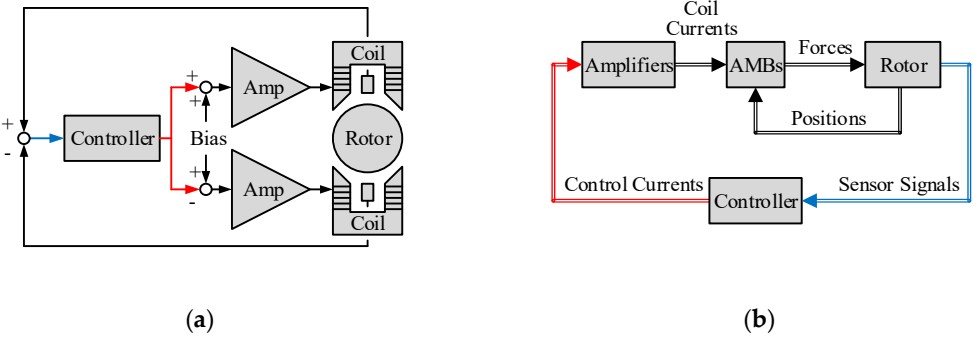

(**a**)  (**b**)

**Figure 1.** Typical AMB control scheme: (**a**) one control axis biasing; and (**b**) MIMO Control block diagram.

By accessing the information available in an AMB, a remote service technician can monitor the condition of the AMB system. Many types of information may be available in a AMB controller, such as rotor speed, hours in operation, temperature, flux, etc., but the signals used for the experimental demonstration in this work were position and current. Figure 2 illustrates the overall concept for the proposed method of AMB condition monitoring via the IoT. In the proposed scheme, the rotor position and the coil current, available from the AMB, are accessed for the IoT. Therefore, a service technician can log in remotely to the gateway and troubleshoot the AMB system. The remote technician is granted the ability to diagnose a variety of equipment malfunctions. The technician might be able to recommend corrective or even preventative action to the AMB end users without the need for an on-site service visit.

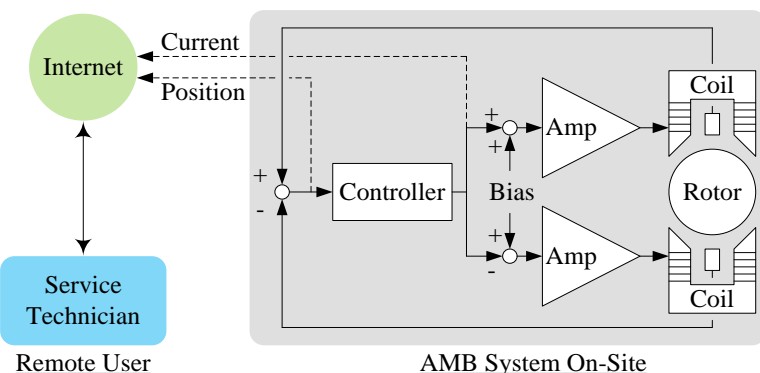

**Figure 2.** Condition monitoring of AMB system via the IoT concept.

For example, if the rotor position is low and/or the coil current is high, shaft overloading may be indicated. The technician can recommend checking the application, or "spec out" a larger AMB. Another scenario could involve the remote technician observing an overly large orbit indicating large unbalance. The technician could recommend rotor balancing before continuing operation.

### 2.2. Experimental System

The experimental system used to develop the proposed IoT condition monitoring solution consists of a stand-alone AMB test rig coupled with an IoT gateway and other hardware. The AMB test rig is model MBC500 by LaunchPoint Technologies, Inc. It has sensor signals that are readily available via a front breakout panel. The IoT gateway selected is the Raspberry Pi 3 Model B single board computer. The experimental system is shown in Figure 3. Figure 3 shows the overall system with: the AMB test rig, added sensor amplifiers and ADC unit on a solderless breadboard, and Raspberry Pi with Ethernet cable to connect to the Internet. The levitated shaft of the test rig has metal collars, which are movable, removable, made of differing materials for differing weights, and may have an adjustable unbalance screw added.

Additional hardware is required for interfacing the Raspberry Pi with the analog signals of the AMB rig. Specifically, an analog-to-digital converter (ADC) board and amplifiers were used to condition the sensor signals to the appropriate range. The following subsections detail the AMB test rig configuration, electronic hardware for IoT implementation, and corresponding software.

### 2.2.1. AMB Test Rig

The MBC500 AMB test rig consists of a single shaft supported by two radial AMBs at either end. Shaft position was monitored by Hall effect sensors to the immediate outside of the magnetic coils. The shaft is stainless steel and 12.5 mm in diameter. Rotation is driven by an air turbine near the inboard AMB.

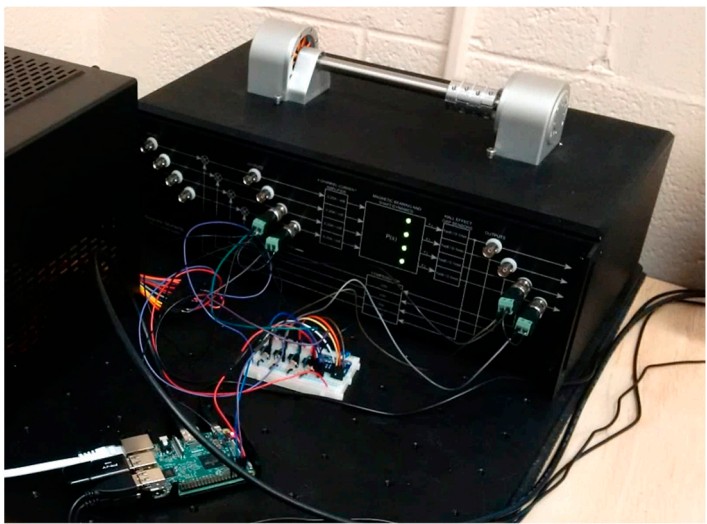

**Figure 3.** Experimental AMB test rig with attached Raspberry Pi single board computer, ADC, and conditioning circuitry.

Movable collars were added to the shaft to create reconfigurable weight and unbalance loads. The collars are approximately 10.5 mm wide and 28 mm in diameter. Two collars (one aluminum and one stainless steel) were used for the present test. The aluminum collar is 15 g and the stainless steel collar is 36 g. The position sensors are 2.8 mm from the ends of the shaft. The locations of the AMBs and the collars on the shaft are shown in Figure 4.

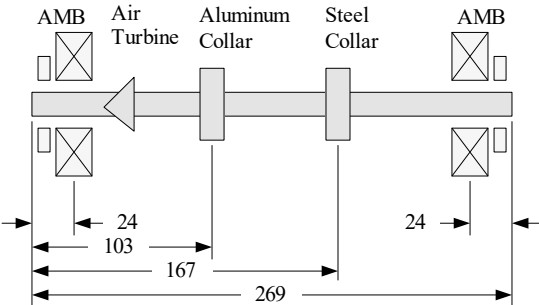

**Figure 4.** Rotor configuration. Dimensions in mm.

The AMBs have eight poles and are wired differentially in the vertical and horizontal axes. Each AMB axis has a bias current of 0.5 A and a nominal gap of 400 μm. The AMB force constant, based on coil geometry, is $2.8 \times 10^{-7}$ N·m$^2$/A$^2$. The current amplifier bandwidth is approximately 720 Hz. The AMB controller built into the MBC500 is local lead-lag type. The built-in controller was used for the IoT condition monitoring study (when the shaft collars were moved).

2.2.2. Hardware Added for IoT

The IoT gateway selected is the Raspberry Pi 3 B (~USD 40). The Raspberry Pi is a single board computer with a 1.2 GHz Broadcom BCM2837 Quad-Core CPU and 1 GB of RAM. It runs the Raspbian operating system, which is Debian based. The Raspberry Pi was selected for this study as it is one of the most readily available IoT gateways. It serves as a cost-effective solution to remote condition monitoring. It is widely obtainable and therefore available to be augmented to older AMB systems already commissioned. In addition, software developed for the Raspbian operating system can be relatively easily ported to other Linux-based systems. A limitation of this IoT gateway solution is it is relatively slow and nonreal-time sampling as a consequence of the operating system. Therefore, it is most useful for monitoring relatively slow rotors or systems with low frequency bearing modes,

subharmonics, external excitations, and substructure modes. The issue of nonreal-time sampling is discussed further in Section 3.

The hardware interface of the Raspberry Pi is general purpose input–output (GPIO) digital pins. To read the analog signals from the AMBs, an ADC must be added. For the current study, a Texas Instruments ADS1115 4-channel 16-bit ADC was utilized (~USD 15 on Adafruit Industries, LLC breakout board). Communication between the Raspberry Pi and the ADC was implemented via standard $I^2C$ digital communication protocol. This protocol requires two wires, one for data transfer and one for a timing trigger. Figure 5a shows the basic scheme for experimental implementation of AMB condition monitoring via IoT.

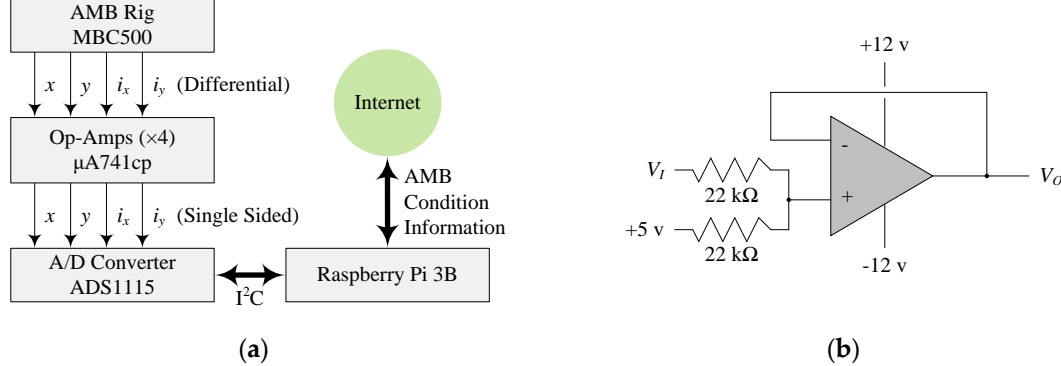

**Figure 5.** Hardware added for IoT implementation: (**a**) overall scheme; and (**b**) op-amp wiring diagram.

The ADC input range is nondifferential, effectively 0–5 V. The AMB sensors' operation range is within ±4 V. The sensor signals were scaled and offset by an array of four summing amplifiers. Texas Instruments μA741cp general purpose operational amplifiers (each < USD 1) were used. The amplifiers were wired as shown in Figure 4b to achieve the usable signal voltage range of 0.5–4.5 V. $V_I$ is the input sensor signal from the AMB and $V_O$ is the output signal sent to the ADC.

A standard PC power supply (~USD 20) provides an economical source of +12 V and −12 V, as well as the 5 V needed to raise to sensor signals. The resistors were balanced to half the overall sensor signal. The exact value of the resistors was selected through trial-and-error for an acceptable impedance match.

One AMB of the fully levitated system was monitored to demonstrate the developed IoT condition monitoring system. The sensor signals monitored were shaft position in the horizontal and vertical directions ($x$ and $y$, respectively) and the corresponding axis coil currents ($i_x$ and $i_y$, respectively). The software loaded on the Raspberry Pi used these data to give insight to the operating conditions of the outboard AMB. The entire assembly of hardware added for IoT is less than USD 100.

2.2.3. IoT Condition Monitoring Software

IoT condition monitoring software was written to collect the output signals of the ADC unit, and to a display a visual representation of the data. This includes rotor position and coil current. The ADC samples each signal, and converts the sampled data from voltage to bits. The bit readout was communicated to the Raspberry Pi via $I^2C$ protocol. The bits were then scaled to recover the real-world signal values. This scaling involved applying an offset and a sensitivity to the vectors of collected data. The offset constants for the positions were obtained (during healthy levitation) by averaging the position vectors. These values were subtracted from all corresponding values in each corresponding vector. The sensitivity, found by comparing a known physical travel to the change in bits, was multiplied into each value in the respective vector. This yielded the calibrated position. A similar process was done for the current, but with the addition of offset to accurately represent the bias current applied.

Next, the vectors of position readings and current readings were plotted to display orbits (*x* vs. *y*) and position *y* with time and frequency. The frequency plot was generated by taking the Fast Fourier Transform (FFT) (via the NumPy library and the command *numpy.fft*) of a reconstructed position vector, as follows: the original sampled position vector had inconsistent time steps because of the nature of the Raspbian operating system. Since the processor was running both the operating system and the monitoring program, the loop running the code may be suspended to maintain the operating system's processes. (The exact sampling rate is discussed in Section 4.) To perform the FFT, which requires a consistent sampling rate, the original data vector and corresponding known time vector were resampled via linear interpolation. Using a resampled vector with 512 elements from the original approximately 400 over 5 s time history was selected. This number of virtual samples was selected to optimize the FFT algorithm while being similar to the actual number of data taken (to maintain fidelity).

The plotting of position and current in the time domain and in the frequency domain gave insight into the AMB system's operating condition. The software placed these figures, as well as the average value of each of the considered parameters after each instance, into a graphical user interface (GUI). The GUI was generated using the matplotlib library and the command *matplotlib.pyplot.plot*. This allowed taking the pre-allocated position and current vectors and expressing them visually. From the Raspberry Pi, a remote monitoring technician could observe the motions of the shaft within the bearing, and alert on-site operating technicians of a possible malfunction.

To connect remotely to the IoT AMB, Secure Shell (SSH), a cryptographic network protocol was enabled to allow a connection from an outside source. The Raspberry Pi hosted a server using the commercially available program *VNC Server*. The technician uses the corresponding program *VNC Viewer* (client) (both by RealVNC Ltd.) to facilitate the connection with a remote framebuffer (RFB) protocol. Similar IoT solutions utilizing VNC Server have been implemented in [36–38]. In general, an RFB protocol transmits screen pixels from one computer (over a network) to another and can also send control events, (e.g., mouse, keyboard, touch screen, etc.) in return [39]. For the current study, the RFB allowed the remote user to activate the IoT program as well as observe the operation of the bearing through the GUI. Therefore, not all data vectors for position, current, etc., need to be transmitted. The program performed data collection and displayed results for a set 5 s time interval. Future editions may allow the program to display latency, constantly update values and automatically replot figures.

## 3. Results

Six trials were conducted to demonstrate the condition monitoring capabilities of the developed AMB IoT system. For each trial, the shaft collars were adjusted to create varying load conditions. Two trials were conducted with the shaft levitated, but not rotating. Two trials were conducted with the shaft rotating. Two trials were conducted with the shaft rotating with added unbalance. For each case, the GUI used by the remote service technician is presented to illustrate how the condition of the AMB system is monitored.

### 3.1. Nonrotating Tests

Figures 6 and 7 show the GUI that a remote service technician would see when executing the IoT AMB monitoring software. The software was executed by calling VNC viewer on the local machine, connecting to the specific Raspberry Pi IoT gateway, and remotely calling the condition monitoring GUI program on the Raspberry Pi. The top of the GUI is a header that displays basic information. (The time period over which data are collected is displayed, in this case 5 s.) There is a blank expansion field for latency to be displayed by a later version of the software. The average values for position and current in the vertical and horizontal AMB axes are also displayed.

The two left plots are orbits, plotting data from vertical vs. horizontal AMB axes. The first plot displays rotor position and the second displays top coil current calculated from the recorded control current. The default scale for the position orbit is half the nominal AMB airgap of $\pm 200$ μm. The default

scale serves as a limit for safe operation predetermined by expert users. Therefore, the technician can easily determine if the rotor is near the limit of safe operation if it nears one of the axes. The default scale for the current orbit is 0–1 A. A non-levitated rotor would sit at (0, −400) µm in the position orbit and (0,0) A in the current orbit.

The center right plot displays the time response of the rotor vertical position over the entire 5 s time history. The right plot displays the corresponding frequency spectrum found with an FFT of the resampled position data.

Figure 6 shows the results for the nonrotating shaft without added collars. The shaft levitated steadily near (0,0) µm, which is the center of the AMB. The coil current was near the bias current, 0.5 A. The calibration of the IoT condition monitoring system can differ from that of the AMB controller.

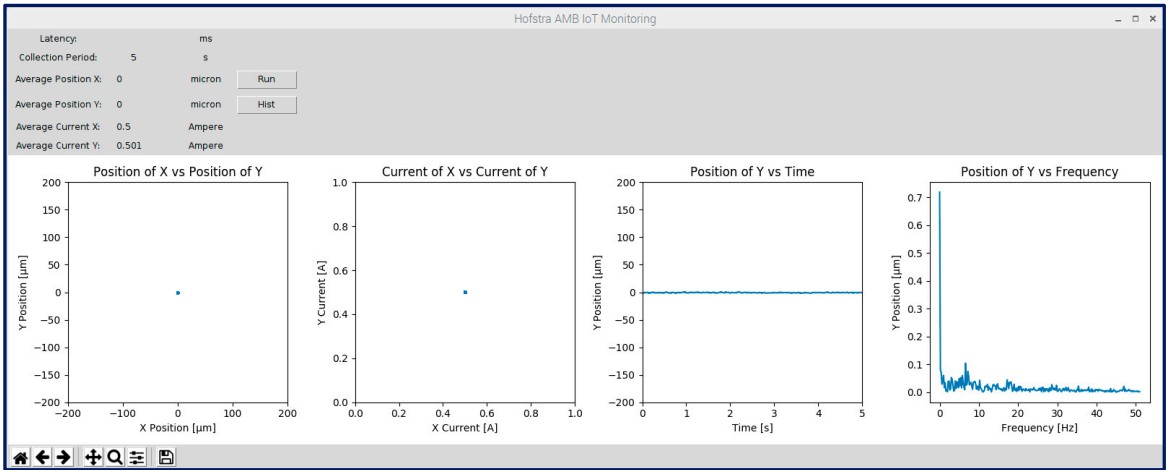

**Figure 6.** AMB IoT condition monitoring GUI display for non-rotating bare shaft.

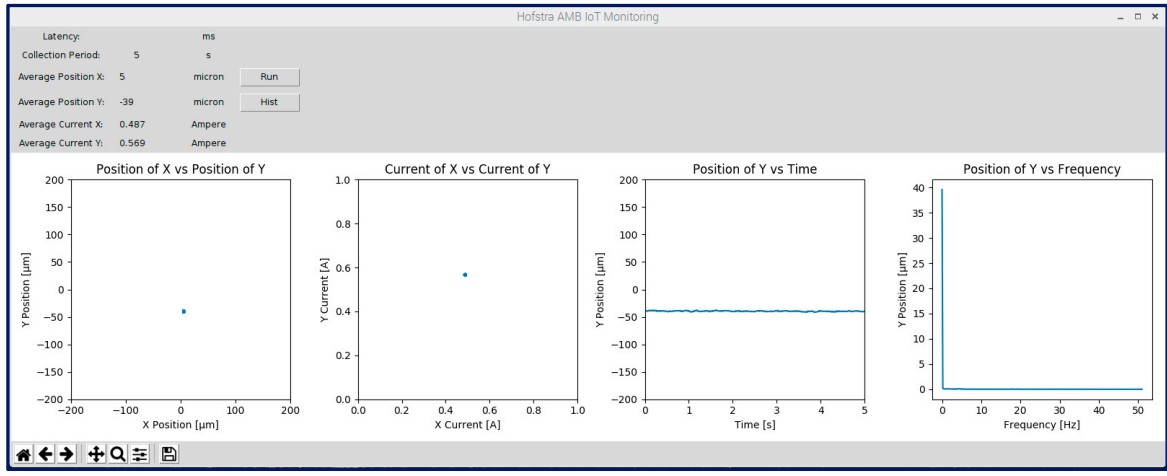

**Figure 7.** AMB IoT condition monitoring GUI display for non-rotating shaft with two collars.

The frequency spectrum depicted in the figure shows only a 0 Hz component for static offset. Note that the AMB controller has no integral action (as in a common PID controller). Figure 7 is for the nonrotating shaft with the two added collars. The remote service technician can infer the static loading condition of the levitated rotor by noting the lower levitated position and increased static current. In the event that the static deflection was too low or the static current was too high, the remote service technician can diagnose rotor over loading and recommend proper corrective action to on-site personnel without the need for an in-person inspection.

### 3.2. Balanced Rotating Tests

The rotor was rotated at 1200 RPM with and without the shaft collars. Figure 8 shows the IoT condition monitoring GUI for the case without the collars and Figure 9 for the case with the collars. The remote service technician can observe the orbit of the rotor inside the AMB air gap caused by the rotation. For both cases, the orbit was consistent and stable. The added weight of the shaft collars caused a static deflection downwards, and a corresponding increase of current (as with the nonrotating cases). The increase of gravity preloading also caused a slight bearing stiffness anisotropy, which led to vertical elongation of the orbit, which was observable by the remote service technician.

The rotation condition of the rotor was further observable in the time plot that displays a consistent harmonic wave. (The frequency spectrum had a peak at approximately 20 Hz, indicating the running speed.) The case with the shaft collars had a slightly higher peak at the running speed because of residual imbalance of the collars. The remote service technician can inspect the frequency spectrum for other components. For example, rotation off of the bearing centerline led to the appearance of a 2× rotation component at 40 Hz, as shown in Figure 9.

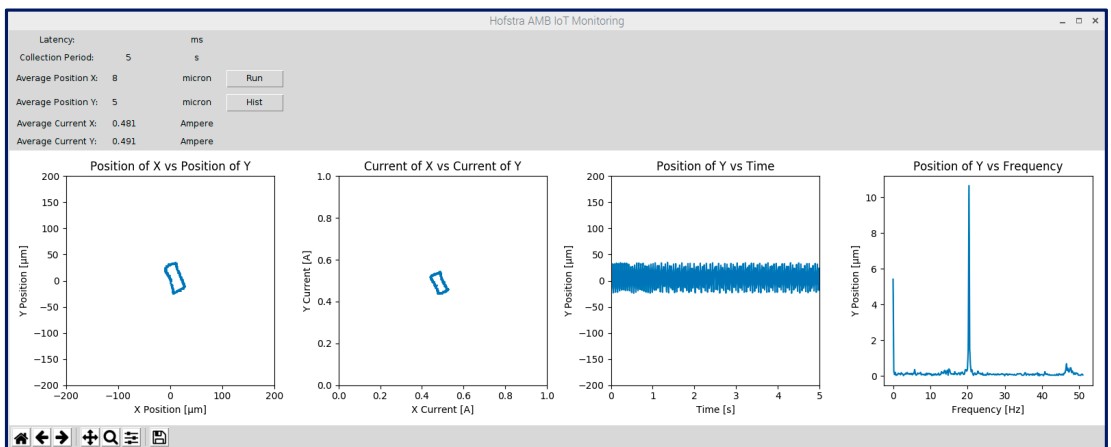

**Figure 8.** AMB IoT condition monitoring GUI display for rotating bare shaft.

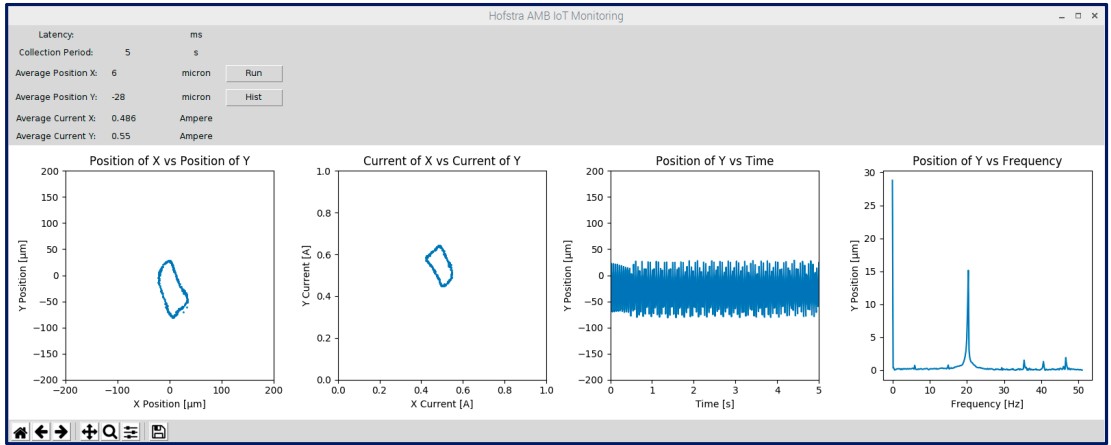

**Figure 9.** AMB IoT condition monitoring GUI display for rotating shaft with two balanced collars.

### 3.3. Unbalanced Rotating Tests

To induce a rotordynamic malfunction, unbalance masses were added to each shaft collar (in the form of a machine screw with exposed head). The resulting unbalance was approximately 6.5 g-mm per collar. Two unbalance tests were conducted. The first had both unbalance screws in the same direction on the rotor to create a static unbalance. The second had the unbalance screws in the opposite directions on the rotor to create a dynamic unbalance. Again, the shaft was rotated at 1200 RPM.

Figure 10 presents the IoT condition monitoring GUI for the static unbalance test and Figure 11 for the dynamic unbalance test. Observing the GUI in Figure 10, a remote service technician can diagnose the rotor unbalance from the slightly increased level of vibrations. This was seen in orbit size, vibration amplitude in time, and $1\times$ frequency peak. The slight increase in level of vibration can alert the remote service technician to the added unbalance.

The dynamic unbalance test shows that the unbalance increased in the aluminum (inboard) collar but the added mass of the stainless steel (outboard) collar countered its own residual unbalance. Therefore, the remote service technician would observe a healthier orbit size, albeit lower, in the bearing gap. The ability to remotely access these data enables the remote service technician to recommend rotor balancing to the end user.

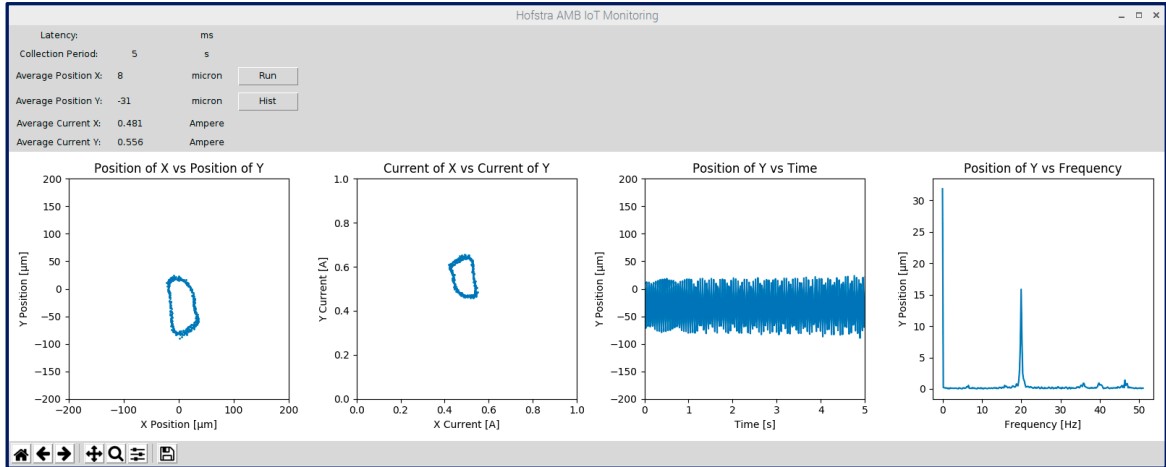

**Figure 10.** AMB IoT condition monitoring GUI display for rotating shaft with two collars and static unbalance.

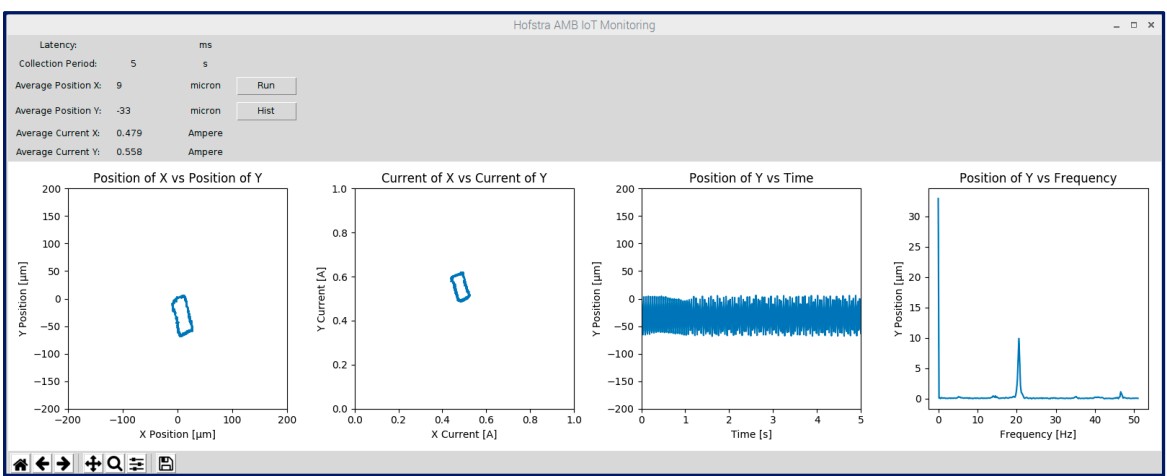

**Figure 11.** AMB IoT condition monitoring GUI display for rotating shaft with two collars and dynamic unbalance.

## 4. Discussion

A limitation of the developed IoT condition monitoring solution is the inconsistent sampling rate that stems from the operating system of the IoT gateway device. This is different from, for example, a dedicated microcontroller that has no operating system and no related background activities. The developed IoT program executed in Raspbian achieved a typical sampling rate of 100 Hz. However, it suffered from periodic delays created as the operating system performs background processes. These are the functions maintaining the operating system and functionality of peripherals and other programs

run by the remote user. Figure 12a shows the time stepping history of a characteristic 5 s condition monitoring run.

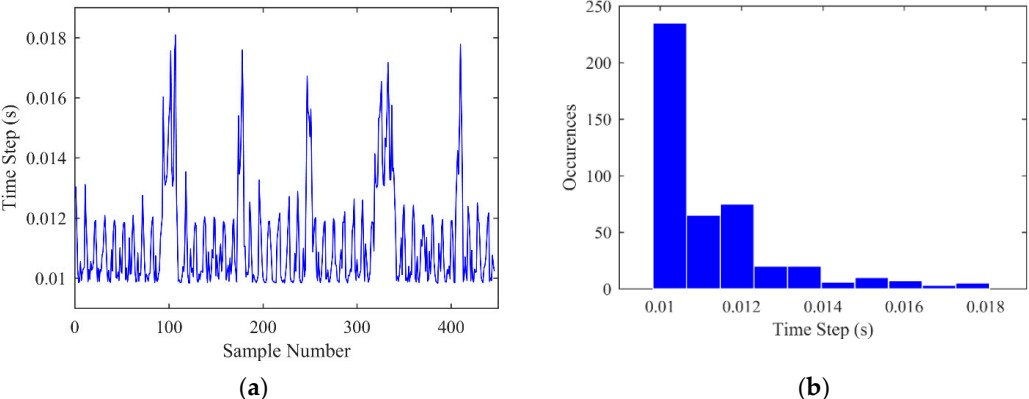

**Figure 12.** Characteristic sampling times for 5 s of IoT data: (**a**) time history; and (**b**) histogram.

The nominal sampling time of 0.01 s is presented as the baseline level in the figure. Frequently, the time was delayed to around 0.012 s. In addition, the data collection was pseudo-periodically delayed even further for several time steps, about every 1 s. This led to a time step as high as 0.018 s.

Figure 12b shows a histogram of the sampling time for this 5 s run. The histogram confirmed that the nominal sampling time was dominant, but interrupted by occasional delays. The current study overcame this limitation by visual inspection of orbits (which are not highly time dependent), and resampling to achieve a practical frequency spectrum. (However, this issue is important in further development of IoT for AMBs, i.e., execution of active control online.) A possible solution might be implementation of a real-time operating system. Another solution might be implementation of a programmable real-time unit on a single board computer.

## 5. Conclusions

This study addressed the problem of remote condition monitoring of AMBs. A solution was proposed to use off-the-shelf IoT hardware and custom software to tie into an AMB's position and current signals. This allowed an OEM technician to observe the signals remotely. The proposed strategy was demonstrated on an AMB test rig. A Raspberry Pi gateway and VNC Server software were used to implement IoT connection. The IoT gateway and other associated hardware cost less than USD 100. Static loading and static and dynamic unbalances were imposed on the experimental rotor. For each case, the conditions of the AMB system were successfully monitored remotely.

Therefore, it was concluded that off-the-shelf IoT hardware and custom software is economical and effective for remote AMB condition monitoring. AMB OEMs can implement similar methods to remotely monitor their products, which are operating on-site for their clients, the end users. This ability will alleviate the need for on-site service calls, and prevent AMB down time.

There are several promising directions for further development of AMBs and IoT. First, cybersecurity should be considered. In other words, mechanisms need to be developed to ensure only intended users can log in and access AMB data. In addition, a mobile application can be developed with which AMB users can check on the condition of the system from arbitrary locations. More complicated is the potential improvement of the IoT scheme for real-time use. Two possible solutions are a real-time operating system for the IoT gateway and using a real-time programmable unit, which would increase hardware cost. Real-time execution will lead to the next stage of development, a cyber physical system (i.e., the feedback control for the AMB will be done through the IoT, making a system in which the real-world dynamics of the system are dependent on the cyberworld). Then, an OEM service technician would not only be able to monitor the condition of an AMB system and diagnose problems, but also might be able to fix problems by changing the control law.

**Author Contributions:** The overall AMB condition monitoring scheme was devised by A.H.P. the coding for IoT implementation of the digital hardware was conducted by P.N.S. The authors worked together to perform the experiment.

**Conflicts of Interest:** The authors declare no conflicts of interest.

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
