# Peer review of "Condition Monitoring of Active Magnetic Bearings on the Internet of Thingsâ€"

_actuators, doi:10.3390/act8010017_

Round 1
Reviewer 1 Report
The manuscript is about the application of the internet of the things to active magnetic bearing systems. No doubts that the topic of IOT and its application is relevant and up-to-date, nevertheless it is not very clear what are the benefits of the proposed solution relative to existing literature that already show examples of AMB systems interfaced by means of the internet. Another issue that arises clear from the manuscript is the non realtime data processing capabilities of the adopted hardware platform. Although this limitation is pointed out at the end of the manuscript, a discussion about the hardware selection should be added at the beginning.
Other comments are reported in the following to motivate the need of an in deep revision.
Introduction:
The part dedicated to the AMB technology is a sort of generic review that is quite well known in technical environment. In my opinion could be made much shorter.
apart from the use of an IOT based platform for data exchange, what is the specific contribution of the manuscript relative to the literature? What IOT technology allows that TCP/IP doesn’t?
Page 3 line 118 - 119 the electric motor is usually not considered as part of the magnetic bearing system. Therefore in fully active AMB systems the number of actuation axes is 5.
Figure 2 and relative text. The information about current and position is just part of the information available in a typical AMB system that could be remotely exchanged.
Figure 4 and text. It is not clear from the text if the shaft is controlled by an axial AMB, that is not shown in the figure.
Figure 5 the output voltage Vo from the buffer circuit is from 1 V to 9 V that seems not consistent with the 0-5V A/D converter range.
Section 2.3.3 is the time corresponding to each sampling known? Without that resampling will not be possible.
It is not clear if all data are sent through the internet or they are locally buffered and just some of them are sent.
Section 4. a low sampling rate of 100 Hz limits the applicability of the proposed solution to low speed rotors. This is not consistent with the high speed application that typical of AMB systems. As pointed out in the section other kind of implementations could avoid this bottleneck by means, for example of intrinsically realtime data processing units. The question arises then about the reason why to adopt a platform with clear intrinsic limitations. This discussion should be added to the manuscript to justify something clear.
Author Response
We thank the reviewer for the many insightful comments and have endeavored to improve the submission as suggested.
The manuscript has been fixed to make clear that the proposed solution uses off-the-shelf IoT equipment and is therefore cost-effective. This is made clear by changes to the abstract, literature review (which more clearly explains the published method currently used in industry), the hardware section (which now reports cost information), and conclusion.
Additional discussion has been added to address the hardware selection and sampling time limitations.
The broad discussion of AMBs was not present in the original conference paper which was written for an audience of AMB experts. We feel it is appropriate to include the introductory material for the wider audience of the journal. It also goes to the editor’s publishing requirements of new material to make a unique document. We have however tried to pare down the basic introduction of AMB to be more concise. This includes discussion in the Introduction and in the Materials and Methods section. In making the discussion more concise, we also removed some issues which were unclear or poorly discussed.
The voltage from the op-amp circuit is 0.5 to 4.5 v. The presence of the matched resistors effectively cut the input voltages in half.
The time of each sample is known. The data processing is done on the local machine and only processed image information is sent through the internet. The text has been improved to make this clear.
Additional discussion of the low sampling rate and hardware selection has been added.
Reviewer 2 Report
Please see the specific comments in the pdf file.

Author Response
We thank the reviewer for the many insightful comments and have endeavored to improve the submission as suggested.
The claim is that off-the-shelf IoT technology can be cost-effective for remote AMB condition monitoring. The key result is that the off-the-shelf IoT technology for remote condition monitoring of AMBs is feasible and low-cost. The abstract has been fixed to make this clear.
The broad discussion of AMBs was not present in the original conference paper which was written for an audience of AMB experts. We feel it is appropriate to include the introductory material for the wider audience of the journal. It also goes to the editor’s publishing requirements of new material to make a unique document. We have however tried to pare down the basic introduction of AMB to be more concise. This includes discussion in the Introduction and in the Materials and Methods section. In making the discussion more concise, we also removed some issues which were unclear or poorly discussed.
The literature review has been improved with more insightful discussion of current state-of-the-art and more clear explanation on the differences of the proposed method. Also, more pertinent references have been added.
Figure 1 has been replaced with more high-quality version.
We leave mobile app for future work.
Additional details on the FFT and GUI codes are reported.
We performed an in situ sensor calibration before the data collection using a dial gage indicator positioned on the shaft near the bearings and taking sensor readout in bits on the Raspberry Pi. The sensor calibration is not perfect but it is reasonably close. The orbit is definitely not beautiful but we believe it is close to what is actually happening using the built in AMB controller.
The conclusion is improved with the cost effectiveness of the developed method.
Round 2
Reviewer 1 Report
Although a rebuttal letter was not included in the new version, the most relevant comments about the previous version have been effectively addressed. For this reason my suggestion is that the manuscript can be published as is.